# Monitoring of Carbon Stocks in Pastures in the Savannas of Brazil through Ecosystem Modeling on a Regional Scale

**Claudinei Oliveira dos Santos** [1,*], **Alexandre de Siqueira Pinto** [2] , **Janete Rego da Silva** [3], **Leandro Leal Parente** [1], **Vinícius Vieira Mesquita** [1], **Maiara Pedral dos Santos** [2] and **Laerte Guimaraes Ferreira** [1]

1   Image Processing and GIS Laboratory (LAPIG), Institute of Socio-Environmental Studies (IESA), Federal University of Goiás (UFG), Goiânia 74001-970, GO, Brazil
2   Ecology Department, Federal University of Sergipe, Aracaju 49060-108, SE, Brazil
3   Tourism and Patrimony, State University of Goiás, Goiás 76600-000, GO, Brazil
*   Correspondence: claudineisantos@discente.ufg.br

**Abstract:** In 2020, Brazil was the seventh largest emitter of GHG (greenhouse gases), releasing ~2.16 GtCO2e (gigatons of carbon dioxide equivalent) into the atmosphere. Activities related to land use contributed approximately 73% of national emissions in that year. Considering that pastures represent the primary land use in the country, occupying approximately 20% of the territory, the mapping and monitoring of C stocks in these areas is essential to determine their contribution to national emissions. In this study, based on the integrated use of the CENTURY model, georeferenced databases, and the R environment, we mapped and analyzed, for the first time, the C stocks dynamics associated with the pasture areas of the Cerrado biome between 2000 and 2019. The average C stocks in the soil (0–20 cm) and in the aboveground biomass estimated by modeling were ~31 MgC·ha$^{-1}$ and ~4 MgC·ha$^{-1}$, respectively, values close to those observed in the literature for the region. Furthermore, the model results corresponded to the edaphic patterns of the region, with the highest average estimated C stocks in Cambisols (~34 MgC·ha$^{-1}$) and the lowest in Neosols (~29 MgC·ha$^{-1}$). The temporal dynamics of soil C stocks in these areas are directly related to the age of the pastures. In fact, stocks tend to be reduced in recently converted areas and stabilized in areas that have been under this land use for a longer time (≥30 years). As a result, a loss of ~103 MtC (millions of tons of carbon) was estimated in the Cerrado pasture soils in twenty years. The mapping and monitoring of C stocks in this land use type through approaches such as the one presented in this study is essential to support the Brazilian government's efforts to mitigate C emissions.

**Keywords:** pastureland; soil organic carbon; ecosystem modeling; Brazilian savanna; CENTURY model; livestock; land use



## 1. Introduction

A major challenge for humanity in this century is to promote technical, social, and political advances in order to keep global warming below 2 °C in relation to preindustrial levels [1]. The UNFCCC (United Nations Framework Convention on Climate Change) was created in 1994 to face this challenge (unfccc.int). An essential objective of the UNFCCC is to reduce the contribution of human actions to climate change, in which carbon (C) is of great importance. In 2016, for example, carbon dioxide ($CO_2$) accounted for nearly 75% of greenhouse gas (GHG) emissions. Therefore, it is crucial to understand the processes related to carbon balance in agroecosystems, which makes it possible to determine the potential of land use in C emissions mitigation [2].

The global carbon cycle involves exchanging this element (i.e., C) between macro-reservoirs (atmosphere, terrestrial ecosystems—plant and soil biomass, ocean ecosystems, etc.) mediated by natural and anthropic processes [3]. In the natural phase, the emission and absorption of carbon may fluctuate over time due to variations in the climate; however,

they tend to balance [3]. Human activities, in turn, have altered this balance between carbon sink and sources; e.g., in the last decade, only 53.4% of average annual emissions (approximately 11 GtC) were offset [3].

Terrestrial ecosystems are crucial to the carbon budget. While human activities (e.g., agriculture, forestry, and other land uses) are important sources of $CO_2$ to the atmosphere, they also result in C sequestration through reforestation or the adoption of optimized management for productivity improvement in cultivated areas [2,4]. In Brazil, the seventh largest emitter of GHGs in 2020, rural activity accounted for ~73% of emissions, with 998 MtCO2e (million tons of $CO_2$ equivalent) attributed to land use changes, mostly deforestation, and 577 MtCO2e attributed to agriculture in 2020 [5]. Pastures occupy about 20% of Brazilian territory, and most of these are degraded or not very productive. Therefore, the pasture reformation promotes productivity gains and, consequently, increases the soil carbon stock, contributing to the mitigation of C emissions in the country [6–9]. Thus, it is necessary to use efficient tools that allow mapping and monitoring of C stocks over time in pasture areas.

A process-based model with computational tools fed with georeferenced environmental databases is a viable and feasible approach for mapping and monitoring C stock in large areas [10]. The CENTURY model (soil organic matter model environment) has been highlighted among several other models (e.g., RothC, DNDC, ParSim) used to simulate C dynamics in land use and land cover scenarios [11]. Initially developed to simulate soil C stocks in pasture areas in temperate regions [12], the CENTURY model has been used to simulate C stocks and greenhouse gas mitigation potential through sequestration of this element in tropical regions [10,13–17].

The CENTURY model was developed in Fortran computational language, and its operation was designed for point simulations and not for automated and efficient processes for large volumes of data (i.e., thousands of points). As a result, application of this model over large extents depends on automating processes through integration with other programming languages and more efficient processing routines. Some initiatives were developed using the CENTURY model on a regional and global scale [10]; however, the tools produced in these studies are either not very accessible or complex to use. Thus, in pursuit of efficient modeling at the landscape scale, we used the R environment to integrate the CENTURY model with a geographic information system (GIS) (codes available at: github.com/claudineisan/rcentury, accessed on 2 November 2022). Based on this tool, as well as the annual land cover and land use data made available by the MapBiomas initiative [18,19], we generated, in an unprecedented and innovative way, the first time series (2000–2019) of carbon stock maps (at 1 km spatial resolution) for the pasture areas in the Brazilian Cerrado biome. In this study, in addition to describing our novel approaches and tools, we analyze the below- and aboveground C stocks dynamics according to distinct pasture ages and soil types.

## 2. Materials and Methods

### 2.1. Study Area

Approximately $\frac{3}{4}$ of cultivated areas in Brazil are occupied by pastures, amounting to ~178 Mha $\pm$ 2.5% in 2017 [18,19]. About 35% of these pasture areas (~62 Mha) are in the Brazilian savannas, locally known as Cerrado (Figure 1) [18]. The Cerrado is the second largest biome in South America, occupying an area of approximately 2 million $km^2$, with climatic patterns characterized by strong seasonality, with well-defined dry and rainy seasons. The average annual temperature ranges from 18 °C to 28 °C, and rainfall from 800 to 2000 mm, with a very strong dry period around April to September (winter season) [20]. The four main soil groups occupy ~86.4% of the entire area of the biome, with Latosol (40.8%) being the most representative class, followed by Neosol (23.4%), Argisol (12.0%), and Plintosol (10.2%) (Available at: geoportal.cprm.gov.br/pronasolos, accessed on 2 November 2022).

## 2.2. CENTURY Model

The ecosystem process-based model CENTURY [12], version 4.5, was used to estimate C stocks in pastures in the Cerrado biome. The procedure involved three steps: (1) adjusting parameters to reduce differences between reference and simulated values; (2) utilization of these parameters to simulate C stocks in an independent dataset, a process also known as validation; and (3) application of adjustment parameters in estimating C stocks across the area classified as pastures in the Cerrado in 2017 [18].

## 2.3. Sites to Model Calibration and Validation

The data used in the model calibration and validation process were provided by researchers who studied the effect of land use on soil carbon stocks in various regions of Brazil [21]. The sites used were distributed along the Cerrado biome, covering the states of Goiás, Mato Grosso, Tocantins, Maranhão, Bahia, and Piauí (Figure 1), with field sampling carried out in 2010.

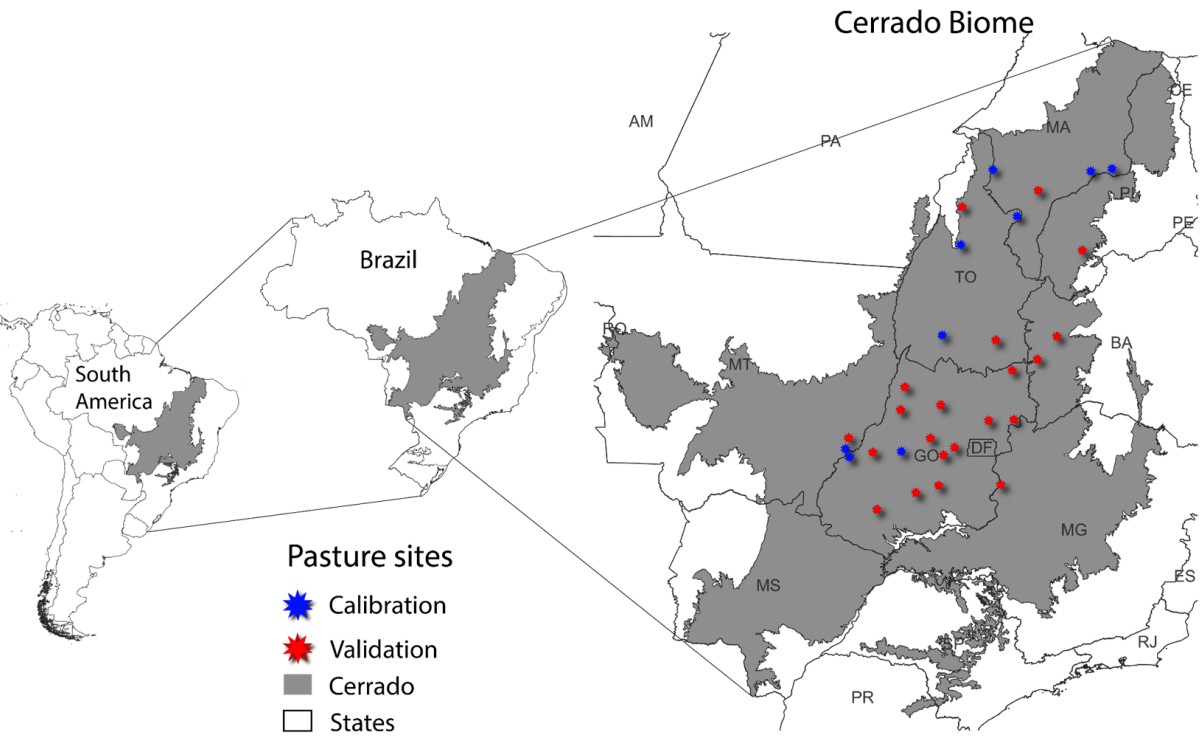

**Figure 1.** Map of the study area (Cerrado biome), with the location of the calibration and validation sites used to simulate carbon dynamics in the pastures of the Cerrado biome based on the CENTURY model.

## 2.4. Pasture Ages

The conversion year from native vegetation to pasture was defined by visual inspection of Landsat 5 satellite images from 1985 to 2010, using the TVI tool (Temporal Visual Inspection) (https://github.com/lapig-ufg/tvi, accessed on 2 November 2022). Nine sites were converted to pastures in the period covered by the images. Their age was precisely determined and used in the model parameterization process, starting the simulation of pasture use in the year of conversion to this land use type. The other sites (*n* = 21) were already occupied by pastures before 1985 and were used to validate the parameterization.

The adjusted CENTURY model parameters were used to estimate C stocks for all pasture areas in the Cerrado with a spatial detail of 1 km². We used the annual series of maps of this land use class produced by Parente et al. [18], covering the period from 1985 to 2017 and later extended to 2019 (available on the Brazilian Pasture Atlas platform:

https://atlasdaspastagens.ufg.br, accessed on 2 November 2022), to identify the area and pasture age. As a reference, we adopted the areas classified in this land use cover in 2017, and we considered the first year in which the area was labeled as pasture to start the simulations of C stocks. The reference map of pastures, with a spatial resolution of 30 m, was resampled using the mode criterion. Thus, the 1 km² pixels remained classified as pastures only if more than 50% of their area had been classified as pastures in the reference map, totaling ~50.7 Mha for the entire biome (Figure 2). The same criterion was used to determine the pasture age, taking the year of conversion of the pixel as the one presenting the most extensive area as first-time classification.

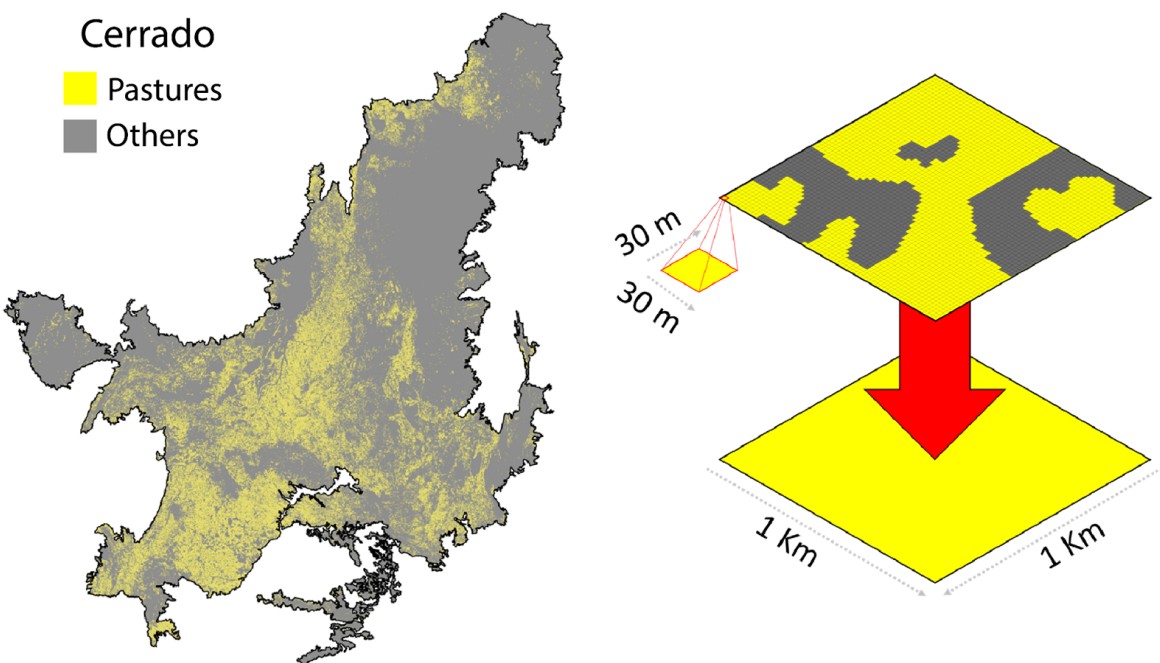

**Figure 2.** Pasture areas in the Cerrado biome resampled at spatial resolution of 30 m × 30 m to 1 km$^{-2}$ were approximately 50.7 Mha in 2017.

*2.5. Weather Data*

For the calibration and validation sites, monthly averages of historical data (1980–2017) of accumulated precipitation (mm) and temperature (minimum and maximum, °C) from the nearest meteorological stations were used. These were obtained through the Meteorological Database (BDMEP) of the Brazilian Institute of Meteorology (INMET). The correction of missing data was performed by linear regression using data from the nearest station, according to the methodology proposed by [22].

For spatialization, the values of monthly accumulated precipitation and temperatures—monthly average maximum and minimum—were obtained from the TerraClimate database [23]. The TerraClimate database has a monthly temporal resolution and a spatial resolution of 4 km, covering the period from 1958 to the present, so provides all the necessary variables for the CENTURY-based modeling process, with the temporal and spatial coverage and frequency compatible with the objectives of this study (see Figure S1). The TerraClimate data, based on models and interpolation, present systematic errors compared to local data obtained from weather stations. Therefore, correction factors were generated for the climatic variables based on data from 30 climatic stations in the Cerrado biome. Through these, the systematic errors of the TerraClimate dataset were corrected as proposed by Carvalho et al. [24]. Reflecting the climatic seasonality of the Cerrado biome, in 2017, the average annual precipitation was 1030 ± 204 mm, while the average monthly maximum temperature was 31 ± 1.4 °C, and the monthly average minimum temperature was 18.9 ± 1.7 °C.

### 2.6. Soil Physicochemical Properties

The edaphic characteristics of the points used in the calibration and validation process are described in Table S1. For each pixel, the model input data relating to soil texture (sand, clay, and silt), pH, and soil density were obtained from the global SoilGrids database [25,26]. At 1 km spatial resolution, SoilGrids provides information on the physicochemical properties of the entire planet in six layers of depth (see Figure S2). The average sand, clay, and silt concentrations were 61.3%, 23.6%, and 15.5%, respectively, while the average pH was 5.1, and the density was approximately 1.4 ton/m$^3$ (Table 1).

**Table 1.** Soil texture, soil density, and pH data in the Cerrado were obtained from the SoilGrids database and used to model carbon stocks in the pasture areas of the biome.

| Variable | Maximum | Minimum | Average | Standard Deviation |
|----------|---------|---------|---------|--------------------|
| Sand (%) | 84.0 | 28.0 | 61.3 | 9.0 |
| Clay (%) | 52.0 | 9.0 | 23.6 | 6.2 |
| Silt (%) | 37.0 | 3.0 | 15.5 | 4.3 |
| pH KCl | 6.5 | 4.3 | 5.1 | 1.6 |
| Density (Kg/m$^3$) | 1535.0 | 1137.0 | 1397.1 | 33.5 |

### 2.7. CENTURY Model Parameterization

The model was run for ten thousand years, using parameters preadjusted by Ferreira [27] to represent the conditions of natural Cerrado vegetation without anthropic changes. This initial simulation aimed to achieve stabilization of C stocks in the soil and vegetation of the studied pasture sites, thus representing a system in balance concerning C inputs and outputs. Cerrado and pasture installation, according to parameter adjustments, were considered in such a way that the simulated soil C stock values could approach the values measured in the field.

The adjustment of parameters related to pasture productivity efficiency (PRDX) was performed based on their effect on C output in biomass and soil, as well as by comparing the values with those obtained in the Rio Vermelho Watershed [28] and in the literature [29]. The allocation of C in roots or shoots of grass, relating to water or nutritional availability, was adjusted using the parameters CFRTCN (1), CFRTCN (2), CFRTCW (1), and CFRTCW (2) to correspond to values found by Oliveira [30]. In addition, we adjusted the parameter value related to the optimal temperature of grass production (PPDF) (Table S2) [31].

The simulated management for pastures was the traditional extensive system practiced in Cerrado, as it is used in most rural properties in the biome [8]. This is characterized by weed control through manual mowing or herbicide, continuous grazing throughout the year, low frequency of pasture fertilization, and cattle herd supplementation. As a result, there is a low stocking rate, depending on the quality and quantity of available grass, and the pastures generally show some degree of degradation [8].

### 2.8. Model Performance Analysis

The performance of the CENTURY model to estimate C stock in soils under pasture in the Cerrado biome was evaluated through the response of the variable SOMSC (carbon in soil organic matter) to the edaphoclimatic characteristics of the site. The mean error of the model was estimated by calculating the root mean square error (RMSE), which considers the differences between the simulated and observed values squared. This result was expressed in proportional terms by dividing the value found by the average of observed soil C stocks (RMSE%). The association between simulated and observed soil C stocks was evaluated using the correlation coefficient (r) described in [11]. The calculation of the Nash-Sutcliffe efficiency coefficient (COE) was also performed, whose values vary between $-\infty$ and 1, where 1 is considered the ideal fit [32]. Another procedure for evaluating the model was to count the number of studied sites whose relative difference between the simulated and

observed value divided by the observed value was within the range of ±25% [33]. Thus, the greater the number of sites within this range, the better the model's performance.

From a regional perspective, the responses of the CENTURY model to soil types in areas covered by pasture in the Cerrado were investigated, having as reference the mapping of the Brazilian soil classes, provided by the Brazilian Institute of Geography and Statistics (available at: geoportal.cprm.gov.br/pronasolos, accessed on 6 November 2022), as well as the age of pastures. In the latter case, soil C stocks were compared between 1985 and 2017. The response of aboveground C biomass to water seasonality was assessed by monitoring the variables AGLIVC (carbon in live aboveground biomass) and STDEDC (standing dead carbon in aboveground biomass) throughout the year 2019.

## 3. Results

### 3.1. Performance Evaluation of the CENTURY Model Per Site

The mean C stocks in simulated pasture soils for the 21 validation sites was $35.2 \pm 6.8$ (mean ± sd) MgC·ha$^{-1}$, similar to the value found in the field by Assad et al. [21] ($35.2 \pm 9.6$ MgC·ha$^{-1}$). The coefficient of efficiency (COE) of the model was 0.58, while the relative mean model error (RMSE%) was 17%. The correlation between observed and simulated values (r), in turn, was significant (0.76; $p \leq 0.01$) (Figure 3A). This correlation coefficient was intermediate compared to the values found by Cerri et al. [6] and Silva-Olaya et al. [34], respectively, of 0.66 and 0.89. In the first study, the CENTURY model was used to study C stocks in pasture areas in the Amazon with different ages and types of management. The second study focused on pastures in the south-central region of Brazil, in the Cerrado and Atlantic Forest biomes.

When considering the model's ability to estimate soil C stocks according to the validation sites, we observed that in 18 of the 21 sites, the relative difference between simulated and observed values was in the range of ±25% (Figure 3B). Therefore, the adjustments made in the CENTURY model to estimate C stocks in soils under Cerrado pasture were satisfactory. Thus, the adjusted parameters were used to build maps based on the model's response to soil and climate conditions in Cerrado pastures obtained from georeferenced databases.

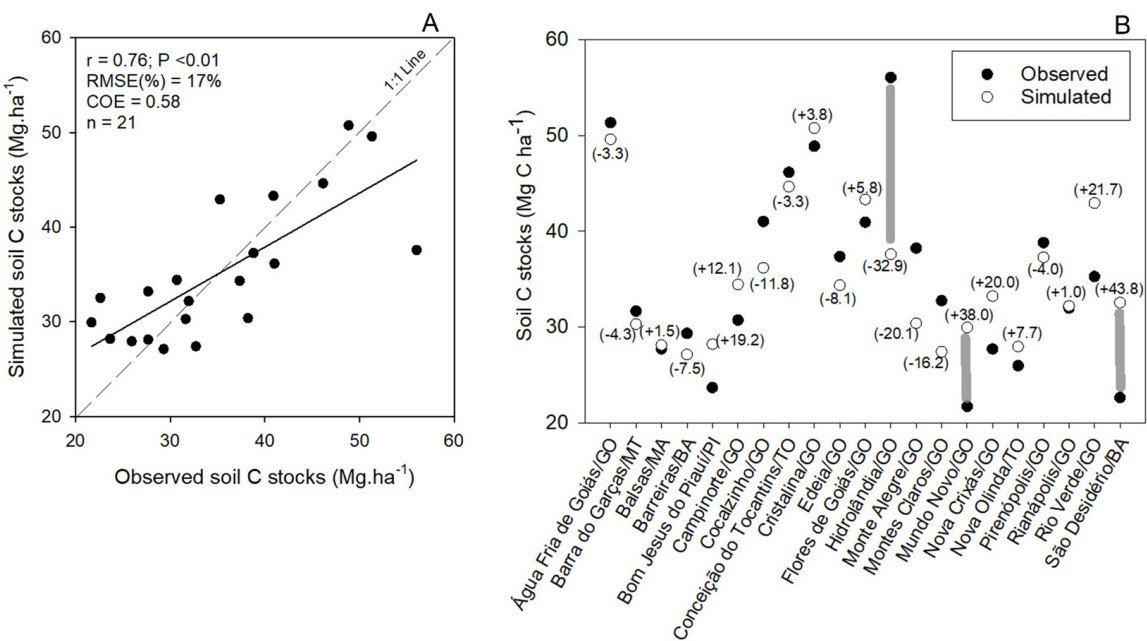

**Figure 3.** (**A**) Relationship between soil C (MgC·ha$^{-1}$) stocks (0–20 cm) observed in the field [21] and simulated by the CENTURY model for pasture areas in the Cerrado biome. (**B**) Soil C stocks (observed and simulated). Values in parentheses are the model error values in percentage. Gray bars indicate sites where errors were outside the ±25% range.

### 3.2. Prediction of C stocks in Cerrado Pastures on a Regional Scale

The C stock (soil + aboveground biomass) in the entire Cerrado area classified as pasture in 2019 was 1.69 PgC, with 89.3% of this value being stored in the soil (1.51 PgC). Fidalgo et al. [35] estimated C stocks in Brazilian soils at ~36.6 PgC in the surface layer (0–30 cm), regardless of vegetation cover. Applying the proportional pasture cover in the Cerrado and the multiplication factor of 0.8 to adjust the C stock value for the soil 0–20 cm layer, as observed by Carvalho et al. [36], would be equivalent to a stock of 1.74 PgC. Therefore, the estimate of C stocks in soils under pasture in the Cerrado, obtained through the CENTURY model, is close to the expected values for the region (13% difference).

C stocks in soil under pasture in the Cerrado (0–20 cm) showed values between ~7.0 and 43.6 MgC·ha$^{-1}$, with an average of ~30.8 MgC·ha$^{-1}$ (Figure 4A). Braz et al. [37] carried out a comprehensive study in pasture areas in the Cerrado, whose clay content varied between 11 and 67%, and the sampling points were classified as productive and degraded. The average soil C stocks (0–30 cm) ranged from 23.9 to 65.9 MgC·ha$^{-1}$, and the general average was 40.6 MgC·ha$^{-1}$. Aboveground C biomass (sum of the variables AGLIVC and STDEDC) was estimated in Cerrado pastures at 0.18 PgC. At the same time, the average was 4.0 MgC·ha$^{-1}$, similar to the stock estimated by Bustamante et al. [7], which was 4.1 MgC·ha$^{-1}$.

Estimated C stocks in the live aboveground biomass ranged from ~1.1 to ~3.2 MgC·ha$^{-1}$, while in the dead component, they ranged from ~0.5 to ~2.1 MgC·ha$^{-1}$ (Figure 4B,C). A similar relationship between the live and dead fractions of aboveground pasture biomass was found in a study carried out in the Rio Vermelho Watershed (GO), where the average C stocks in the live and dead biomass sampled in the field were ~1.6 MgC·ha$^{-1}$ and ~1.1 MgC·ha$^{-1}$ [38].

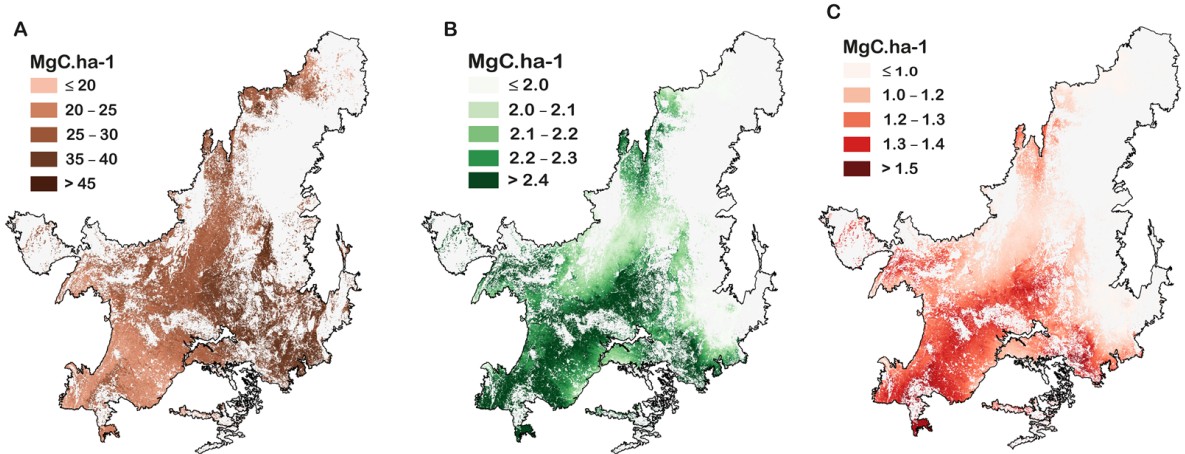

**Figure 4.** Estimated carbon stock for pasture areas of the Cerrado biome in the year 2019. (**A**) Carbon in soil organic matter. (**B**) Carbon in living aboveground biomass. (**C**) Standing dead carbon in aboveground biomass.

The forage quality for animal feed varies throughout the year in response to water seasonality, with better conditions in the rainy season. Consequently, a higher proportion of green biomass is expected from the total biomass available. When considering the dynamics of C stocks in aboveground biomass in 2019, the C ratio in live biomass of total biomass peaked at 84.8%. In the dry season, this proportion reduced to ~40.7% (Figure 5). Because the model can monitor both the dead and live components of aboveground biomass, the results can be compared with satellite data—such as GPP (gross primary productivity) and NDVI (normalized difference vegetation index), used to study the dynamics of biomass and vegetative vigor in pastures—in future studies [39,40].

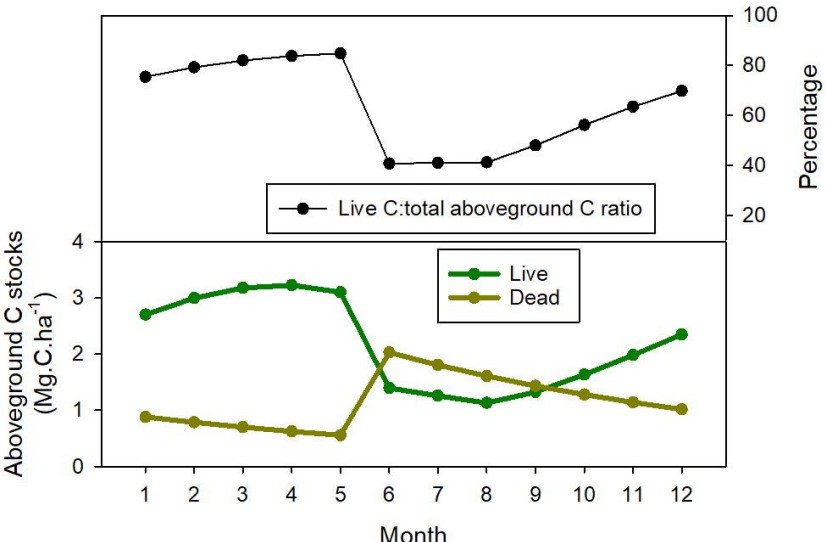

**Figure 5.** Seasonal pattern of estimated carbon stock for live and dead aboveground biomass in grassland areas of the Cerrado biome in 2019.

### 3.3. Effect of Soil Type and Age of Pastures on Carbon Stocks

The model responded to Brazil's different soil types under pasture, with the lowest average C stock in Neosols (~29 MgC·ha$^{-1}$), the soil class corresponding to ~18% of the studied area. The highest stock was found in Cambisols (34.0 MgC·ha$^{-1}$), which occupy approximately 11% of the Cerrado pasture area (Figure 6A). Intermediate values (~31 MgC·ha$^{-1}$) were observed in Argisols, Latosols (higher proportion of land cover, about 48%), and Plintosols (~10% coverage). Gomes et al. [41] estimated carbon stocks in soils in the Brazilian territory (up to 100 cm in depth). They found the ordering of values regarding soil type similar to that observed in the present study, with higher stocks in Cambisols.

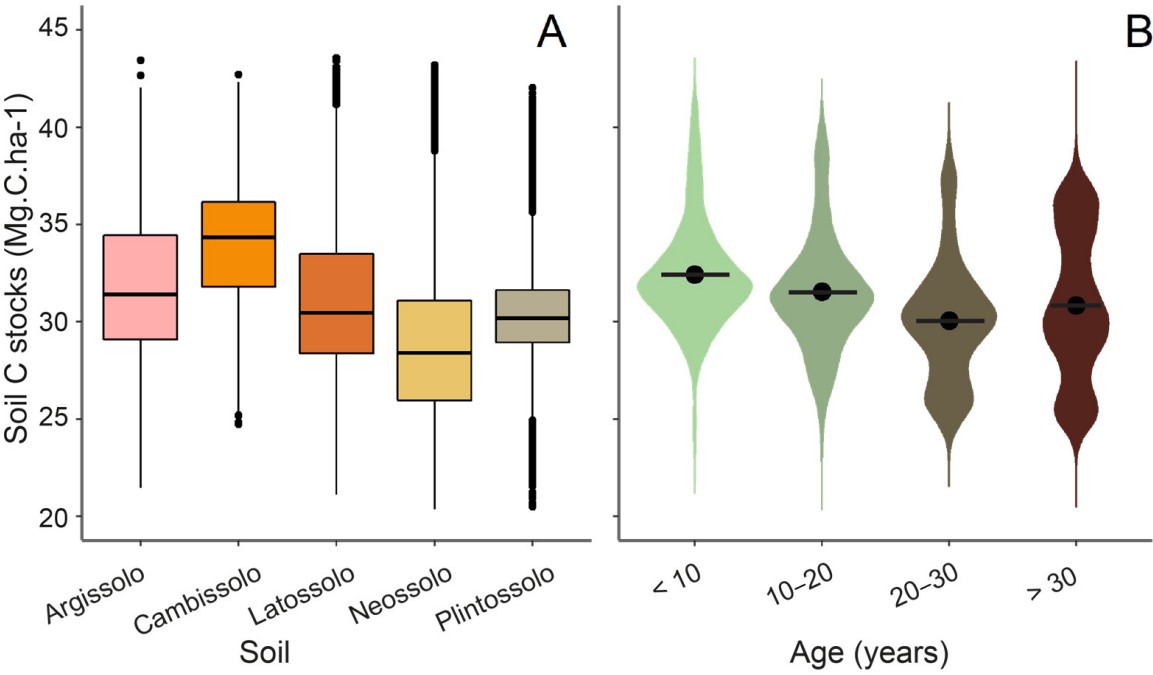

**Figure 6.** Estimated carbon stock for pasture areas of the Cerrado biome in the year 2019. (**A**) Carbon stock according to soil type. (**B**) Carbon stock according to pasture age (*3.0% of the area occupied by pastures in the biome belonged to unanalyzed soil orders).

Another important factor responsible for variations in C stocks in soils is the age of the pasture. In the younger class, the estimated average stock was ~32 MgC·ha$^{-1}$, while soil C stocks were reduced to ~30 MgC·ha$^{-1}$ in pastures up to 30 years old (Figure 6B). The soil C stocks stabilization was reached in older pastures. Corbeels et al. [42] found no significant difference between pasture areas aged 18 years in the first sampling and those resampled after 8 years.

### 3.4. Monitoring C Stocks in Cerrado Pastures

Pasture soils can function as a sink or source of C to the atmosphere, depending on the management practices adopted [6]. The pasture soil C dynamics simulation under traditional management in the Cerrado showed a net loss of C of ~104 MtC between 2000 and 2019. Soil carbon losses are particularly concerning since it is a compartment for long-term C storage. The estimated C loss in Cerrado pastures in the studied period was similar to the emissions (in $CO_2$ eq) of the Brazilian industrial process sector in 2020 (i.e., 99.97 MtC). Furthermore, it corresponded to ~19% of the agricultural sector emissions in the same year (557.02 MtC) [5]. Thus, the results obtained in this study reinforce the need to invest in sustainable pasture management practices to reduce emissions from this type of land use [4,7,8].

We noted that more than half of the soil C loss (~60 MtC) was observed in the first decade of the studied period. Between 2000 and 2009, a larger pasture area was within the highest loss class ($\leq -1.5$ MgC·ha$^{-1}$) in comparison to the following decade (Figure 7). Considering that deforested areas in Brazil are mainly primarily converted to pasture [18], and that these areas are subjected to traditional management, we could expect an increase in recent pasture areas, which are more prone to higher soil C losses as observed in our results. Therefore, this practice must be discouraged because it is against the world's efforts to mitigate carbon emissions from human activities. Furthermore, why does Brazil proceed with deforestation to install new pastures instead of reforming existing pastures? According to Carvalho et al. [43], more than 80% of pasture areas in this country are found in some state of degradation.

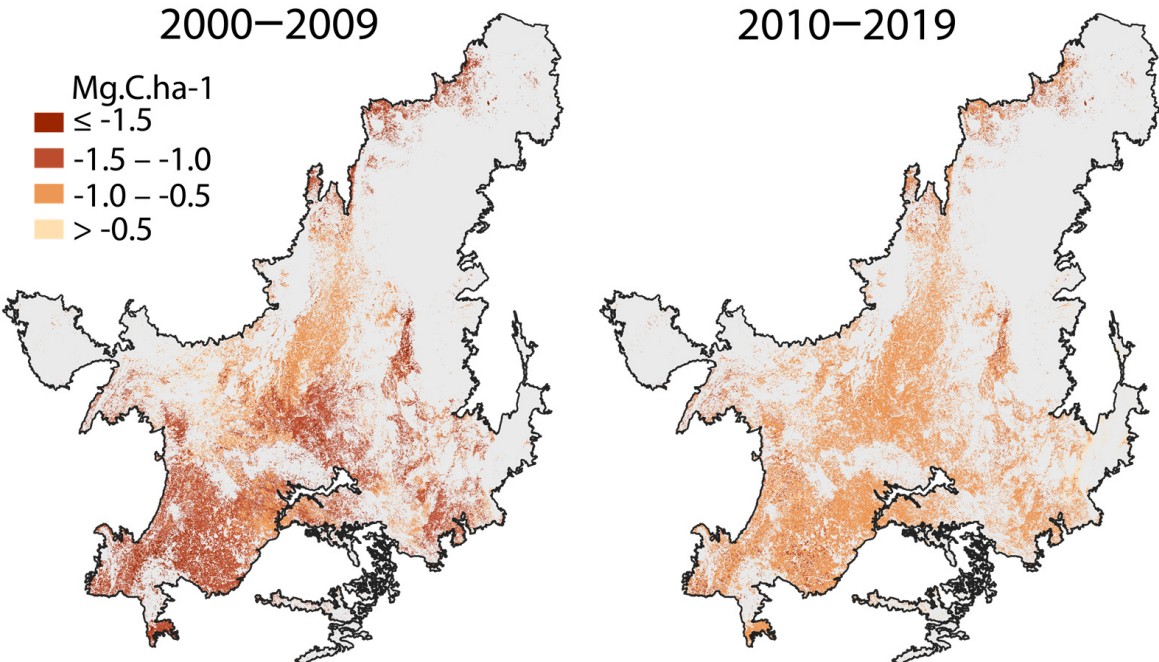

**Figure 7.** Changes in soil carbon stocks in pastures in the Brazilian Savanna (the Cerrado) in two periods: (**Left**) 2000 to 2009 and (**Right**) 2010 to 2019.

In this context, the time series of C maps, such as those presented in this study, is an interesting tool for estimating C fluxes (emissions and absorption) [44]. These datasets are also essential to support the creation and implementation of climate policy initiatives to reduce C emissions in the land use sector [45,46]. They can also inform the government and society about the possible outcomes of these policies, thus assisting in decision making [47].

## 4. Conclusions

This study evaluated the potential of using the CENTURY model integrated with georeferenced edaphoclimatic databases and the R environment to map and monitor C stocks in Cerrado pastures. The results showed that this approach is robust for simulating C stocks on a regional scale. In addition, we observed responses to essential aspects that regulate carbon dynamics in ecosystems, such as climatic factors, pasture age, and soil classes.

Pasture in the Cerrado biome submitted to traditional management was responsible for a soil C loss of ~104 MtC in 20 years. Therefore, Brazilian environmental policies should promote alternative management to this land use, aiming to reduce C emissions and support C sequestration by increasing pasture productivity. This scenario could contribute to the efforts of the Brazilian government to reach the net-zero carbon goals.

The application of the strategies and protocol presented in this study to all the six Brazilian biomes, considering different pasture management traits and edaphoclimatic conditions, could support the national carbon emission inventories and assist in formulating and implementing policies aimed at climate change mitigation.

**Supplementary Materials:** The following supporting information can be downloaded at: https://www.mdpi.com/article/10.3390/land12010060/s1. Figure S1: Maps of climate variables in the Cerrado, obtained from the TerraClimate database and used to model carbon stocks in the pasture areas of the biome. (TMIN—minimum temperature; TMAX—maximum temperature; PPT—precipitation. For illustration purposes, we used data from June/2020 and December/2020). Figure S2: Soil texture maps in the Cerrado, obtained from the SoilGrids database and used to model carbon stocks in the pasture areas of the biome. Density and acidity (pH) maps were obtained on the same basis. Figure S3: Carbon stock mean spatial variation (carbon stock on a pixel basis normalized by the mean Cerrado carbon stock) in the pasture areas of the Cerrado biome (layer 0–20 cm). Table S1: Location and soil characteristics in the 0–20 cm layer of the sites used in the calibration and validation process of the CENTURY model to represent pasture areas in the Cerrado. Table S2: Modified parameters in CENTURY to simulate traditional pasture management in the Cerrado biome. Table S3: Datasets used to model pasture carbon stock in the Cerrado biome pastures.

**Author Contributions:** C.O.d.S., A.d.S.P., L.G.F. and J.R.d.S. conceived the idea and the methodological approach of the study. C.O.d.S., M.P.d.S., L.L.P. and V.V.M. processed the data and, together with L.G.F., analyzed the results. All the authors contributed to the discussions and writing and revision of the manuscript. L.G.F. was responsible for the funding acquisition. All authors have read and agreed to the published version of the manuscript.

**Funding:** This research received no external funding.

**Data Availability Statement:** Not applicable.

**Acknowledgments:** This work, part of the MapBiomas initiative (http://mapbiomas.org accessed on 2 November 2022), was supported by the Gordon and Betty Moore Foundation (GBMF), The Nature Conservancy (TNC), the Goiás State Research Foundation (FAPEG), the Coordination for the Improvement of Higher Education Personnel (CAPES), and the Brazilian Research Council (CNPq).

**Conflicts of Interest:** The authors declare no conflict of interest.

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
