# Peer review of "Monitoring of Carbon Stocks in Pastures in the Savannas of Brazil through Ecosystem Modeling on a Regional Scale"

_land, doi:10.3390/land12010060_

Round 1

Reviewer 1 Report

You presented a very relevant issue in your paper. Also, your paper is very well written in the English language.

However, I have about 15 minor and major revision comments that you may want to attend to and revise the manuscript. Pease find the attached file.  

If any of the comments are not clear, please don't hesitate to contact me through your editor.

Author Response

Response to Reviewer 1 Comments

You presented a very relevant issue in your paper. Also, your paper is very well written in the English language.

However, I have about 15 minor and major revision comments that you may want to attend to and revise the manuscript. Please find the attached file. 

Thank you for your very pertinent comments. All the suggestions were taken into consideration in the thoroughly revision of the manuscript.

  • Line 15. What percentage of pasture land use?

The percentage of pasture area in the country was included in the abstract.

  • Line 24. This sentence is too long.

The text was revised as suggested.

  • Line 35. Please mention the broader problem first before you mention the challenge or difficulty.

We decided not to change this part of the text for style reasons.

  • Line 44. Which element? Please be specific since you are starting a new paragraph

We made the text clearer, as requested.

  • Line 57. Add this statistics to the abstract in the sentence where I asked about the % of pasture land use

This statistics has been added to the abstract, as recommended.

  • Line 77. Has this approach been used in regions where pasture forms about 20% or more of the landscape? Please indicate it. If it has been used before in such regions, please justify why you believe it is appropriate to use in the Brazilian pasture region.

We rephrase this part of the text, as the main objective of this study, as well as its potential for innovation, was not centered on the development of a computational tool per se, but rather on the modeling of carbon stocks at the landscape scale.

  • Line 84. Any map to show your study area, Brazil, and South America? Please add if you have not already added one which shows all the three areas in one map figure.

We moved the figure 1 to this section; and we placed an inset in figure 1 with the purpose of locating the Cerrado biome in relation to the other biomes in the country.

  • Line 106. Please justify why you believe it is still appropriate to use data which is approximately 12 years old to tell a story about a problem in this current time. Here, you may want to provide a justification of your choice of data use, strengths and limitations of using such data.

The adjustments in the model parameters are needed to achieve the minimum possible differences between simulated and observed values. However, as we used a process-based model, it is also necessary to represent the soil carbon dynamics over time. Therefore, the land use simulation was carried out from the pre-use condition, i.e., simulating carbon stocks since native vegetation conditions up to the pasture reference year. Another essential point for adequately representing the variation in soil carbon stocks over time is that we used pasture sites with different ages for model calibration. Combining these efforts enabled us to work with reference data older than those used (or more recent), and such strategy should not affect the model's performance, since its adjustment was guided by the response of the variable under study (soil carbon stocks) to edaphoclimatic conditions (as well as to land management).

In addition, and specifically concerning the pasture field data (from 2010), it is important to emphasize that it was acquired in the period covered by this study (i.e. 2000 to 2019). And this database is the most complete and best spatially distributed available for the Cerrado biome.

Line 126. This sentence is too long and difficult to follow and understand. In the same sentence (1) Why resampling? (2) What are the potential impacts of the resampling on your results?

We made the text clearer, as suggested. Concerning the pasture pixels resampling (from 30m to 1km), this was necessary due to the much coarser spatial resolution of the other datasets utilized in this study (e.g. soil texture attributes at 1km). Concerning the implications of such upscaling, this low resolution is appropriate for assessing carbon stocks at regional and landscape scales, not being appropriate for fine scales such as farmland levels.

  • Line 134. The pasture areas are represented by yellow on the map. How about the dark areas? What do they represent?

The figure was slightly modified; i.e. in the legend we included the class "others", which is associated with the gray color in the map.

  • Line 145. What are the limitations of integrating this 4 km resolution data with the 1 km resolution pasture data?

Environmental data availability is always a critical issue for large-scale studies, particularly when long-time series are involved (as is the case of our study). Within this context, the TerraClimate databases, in spite of their coarse spatial resolution and reanalysis-based retrievals, were for sure the best option available. Nevertheless, and it is important to emphasize, the spatial climate variability in the Brazilian savanna tends to be lower than that observed for the edaphic and land use variability. Therefore, modeling carbon stocks at 1 km spatial resolution was certainly the best possible compromise regarding the input data at distinct spatial resolutions.

  • Line 219. Have you started your discussions already? I thought you will present only the results here and then at the discussion section indicate what the results implies/mean and compare them to previous Additional Comments: Since you do not have a separate discussion section, please name this section as "Results and Discussions". Also, be sure to write what every major finding implies. For the comparison of results to that of other findings in the literature, I think you did an excellent job

The section has been renamed as recommended and the text has been improved to include the implications of major findings.

  • Line 276. This should rather be fixed somewhere in the methods section. I don't remember you mentioned the dead and live biomass in the methods section.

References to live and dead biomass components were included in the methods section as recommended.

  • Line 337. Your conclusions are okay.

Reviewer 2 Report

I appreciate that the authors explained clearly why they use Century model in their study. It would improve their discussion if they mention few more reference research which apply the Century model. Also please add some significant findings of those. 

It would be great if the weather data is represented in a table. 

Please mention the name of the pre-adjusted parameter in section 2.7 Century model parameterization. 

Line 91 under 2. Materials and Methods, please add the appropriate format of reference

Author Response

Response to Reviewer Comments

  • I appreciate that the authors explained clearly why they use the Century model in their study. It would improve their discussion if they mention a few more reference research which apply the Century model. Also please add some significant findings of those.

The discussion was improved as suggested.

  • It would be great if the weather data is represented in a table.

The weather data was spatially presented in maps in the supplementary material. And these maps have been improved, i.e. we considered both  the rainy and dry seasons.

  • Please mention the name of the pre-adjusted parameter in section 2.7 Century model parameterization.

The respective values and meanings of each Century parameters have been included in the Supplementary Material Table S2.

  • Line 91 under 2. Materials and Methods, please add the appropriate format of reference

The reference format was revised.

Reviewer 3 Report

Although this is an interesting study, it could be useful for understanding the C stocks dynamics in the pastures of the Cerrado biome. Please consider following comments to improve the quality of the manuscript:

1.The Century model has been in use for yearsthe authors have developed a tool integrating the Century model with the Geographic Information System (GIS) in the R environment, which allows an efficient landscape scale modeling, How can you demonstrate or show that this tool is efficient, that it is more effective than previous tools, to demonstrate the innovation of the technology?

2. In ” 2.1 Study area”, please add a location map of study area.

3. Weather data (accumulated precipitation and temperature), Soil physicochemical properties data are very important, also should be It should also be visualized in map.

Approximately 3/

4. The CENTURY model used in this study requires a lot of data, and its spatial resolution varies greatly, ranging from 1km, 4km and 30m. Do these differences have an impact on the research results? How does the author consider this issue?

5. As mentioned above, this study uses multi-source data. Please list all the data in a table so that readers can know it at a glance and then judge the reliability of the source and the rationality of use.

6. In line 228-230, the relative difference between simulated and observed values was in the range of ±25%. Why do the author believe that the adjustments made in the Century model to estimate C stocks in soils under Cerrado pasture were satisfactory? Could you explain in more details?

Author Response

Response to Reviewer Comments

Although this is an interesting study, it could be useful for understanding the C stocks dynamics in the pastures of the Cerrado biome. Please consider following comments to improve the quality of the manuscript:

  1. The Century model has been in use for yearsthe authors have developed a tool integrating the Century model with the Geographic Information System (GIS) in the R environment, which allows an efficient landscape scale modeling, How can you demonstrate or show that this tool is efficient, that it is more effective than previous tools, to demonstrate the innovation of the technology?

We developed an integration between Century and GIS via R as a solution to the objective of this study. And as part of our research group philosophy towards open science, we decided to make the developed solution public. But presenting it as a technological innovation wasn't part of the main objective of the study. Therefore, we didn't consider a specific discussion on the efficiency and performance analysis of the tools we developed.

  1. In ” 2.1 Study area”, please add a location map of the study area.

A figure of the study area was included, as recommended.

  1. Weather data (accumulated precipitation and temperature), Soil physicochemical properties data are very important, and should also be visualized in a map.

The supplementary material includes maps of all the edaphoclimatic variables. In addition, a summary of the edaphoclimatic data for the calibration and validation sites have been added to the supplementary material (Table S1).

  1. The CENTURY model used in this study requires a lot of data, and its spatial resolution varies greatly, ranging from 1km, 4km and 30m. Do these differences have an impact on the research results? How does the author consider this issue?

In order to accommodate the variations in dataset resolution, we decided to work with a "compromise" resolution of 1km, which is computational viable. Considering that the climate spatial variability in the Cerrado biome is lower than edaphic variability, the downscaling of these data from 4km to 1km didn't bring limitations to the study.

Specifically concerning the resampling of the pasture area from 30m to 1km, the 1km pixel was assumed as pasture only if more than half of the 30m pixels it comprises had been identified as pasture. Such scaling approaches make our results unsuitable for property level analysis, although they are consistent and appropriate for landscape and regional analysis. 

  1. As mentioned above, this study uses multi-source data. Please list all the data in a table so that readers can know it at a glance and then judge the reliability of the source and the rationality of use.

A table listing and detailing all the datasets used in our study has been included in the supplementary material.

  1. In line 228-230, the relative difference between simulated and observed values was in the range of ±25%. Why do the authors believe that the adjustments made in the Century model to estimate C stocks in soils under Cerrado pasture were satisfactory? Could you explain in more detail?

Carbon stocks in the Cerrado biome pastures present a high spatial variability, reflecting edaphoclimatic conditions and the heterogeneity of production systems. Therefore, the ±25% difference between the modeled and observed carbon stocks were considered satisfactory as a first attempt for large-scale simulations. The reference and justification for using this ±25% range are included in section 2.8 "Model performance analysis" of the study.

Round 2

Reviewer 3 Report

Most of the issues have been properly addressed and questions explained, but the innovation of the research needs to be further highlighted.

Author Response

Response to Reviewer Comments

  1. Most of the issues have been properly addressed and questions explained, but the innovation of the research needs to be further highlighted.

A very bold and original initiative called MapBiomas (mapbiomas.org) is mapping, in an unprecedented way and on an annual basis, different land cover and land use classes for all Brazil (dating back to 1985). And through this time series of maps, publicly available, it was possible to identify the year in which each patch of natural vegetation in the Cerrado biome was converted to pasture.

This precise initial information, crucial for estimates of carbon stocks via ecosystem modeling, is an unique aspect of our study. Specifically, in the present manuscript, we developed an integration between Century and GIS in the R environment, which, in an unprecedented and automated way, feeds the model with information extracted from this series of maps and edaphoclimatic databases. Likewise, all simulations of carbon stocks are carried out automatically.

The R-based tool we developed represents a major advance in the monitoring of carbon stocks in pasture areas, incorporating the space-time dynamics of this land use class. This is the first study to provide a time series of carbon stocks with a spatial resolution of 1km for the Brazilian savannas pastures.

As requested, these aspects, which make our study truly innovative and distinctive, were incorporated into the text.
